# Newborn Screening for Mucopolysaccharidosis I: Moving Forward Learning from Experience

**DOI:** 10.3390/ijns6040091

**Published:** 2020-11-19

**Authors:** Lorne A. Clarke, Patricia Dickson, N. Matthew Ellinwood, Terri L. Klein

**Affiliations:** 1Department of Medical Genetics, B.C. Children’s Hospital Research Institute, University of British Columbia, Vancouver, BC V5Z-4H4, Canada; 2Department of Pediatrics, Washington University School of Medicine, St. Louis, MO 63110, USA; pdickson@wustl.edu; 3National MPS Society, Durham, NC 27707, USA; matthew@mpssociety.org (N.M.E.); terri@mpssociety.org (T.L.K.)

**Keywords:** lysosomal storage disease, mucopolysaccharidoses, glycosaminoglycans, pseudodeficiency

## Abstract

There have been significant advances allowing for the integration of mucopolysaccharidosis I into newborn screening programs. Initial experiences using a single-tier approach for this disorder have highlighted shortcomings that require immediate remediation. The recent evaluation of a second-tier biomarker integrated into the MPS I newborn screening protocol has been demonstrated to greatly improve the precision and predictive value of newborn screening for this disorder. This commentary urges newborn screening programs to learn from these experiences and improve newborn screening for mucopolysaccharidosis I and future mucopolysaccharidoses newborn screening programs by implementation of a second-tier biomarker analyte.

## 1. Introduction

The recommendation by the Federal Advisory Committee on Heritable Disorders in Newborns and Children (ACHDNC) to add MPS I to the Recommended Uniform Screening Panel (RUSP) in the US in 2016 was, for the main part, fueled by two factors; the compelling long-term data showing that the age of initiation of definitive therapy, particularly HSCT in severe MPS I, has a dramatic impact on the ultimate outcome of patients and MPS I registry data showing that in standard medical practice there is considerable delay between symptom onset and diagnosis of MPS I [1,2]. In some cases, the diagnostic odyssey extended over 10 years and the therapeutic odyssey much longer. Other important considerations underlying this recommendation were the limited experiential data from studies of long-term enzyme replacement (ERT) in attenuated MPS I patients and sibling comparative studies indicating that the degree of disease burden at the time of initiation of ERT influences ultimate outcome [3,4,5,6]. Additional studies of ERT and HSCT in animal models of MPS I and other MPS disorders also strongly indicate that reversibility of disease symptoms particularly those in the CNS, skeletal and cardiac tissues is limited, but disease prevention or reduction of the rate of disease progression is possible [7,8,9,10,11,12]. Undoubtedly early initiation of therapy requires early diagnosis. Early symptom recognition by primary health care providers leading to rapid diagnostic confirmation is a fundamental challenge for many rare diseases as the early symptoms exhibited by patients often represent common ailments that have an *a priori* low positive predictive value for diagnosis. In the case of MPS I these include inguinal or abdominal hernia, frequent early ear, nose and throat infections, hydrocephalus and skeletal symptoms including abnormalities in joint range of motion, spinal disease (gibbus) or short stature. Studies performed in the Netherlands in 2018 showed that intensive education and awareness initiatives directed to primary care providers highlighting the early symptoms of MPS and provision of easy access to diagnostic tools or subspecialty services did little from a population perspective to reduce the mean age of diagnosis for MPS patients [13]. In addition, MPS I registry data have shown that although there have been improvements in the time from diagnosis to initiation of therapy particularly for attenuated MPS I patients, there has not been evidence of significant improvements in time to diagnosis [14]. Therefore, newborn screening strategies represent the logical path towards ultimately improving outcomes for MPS I patients.

## 2. Experience

Newborn screening for MPS I has been performed in pilot studies and more recently within regional newborn screening programs in Taiwan, Italy and various states within the USA. Data from these experiences have recently been summarized by Gragnaniello et al. [15]. The initial approach involved a single-tier analysis of iduronidase activity (IDUA) on standard blood spot cards by either tandem mass spectrometry or digital microfluidics fluorometry. Newborns with low IDUA activity were subsequently referred for blood based IDUA activity confirmation and/or specialized clinical evaluation. The threshold for “low IDUA activity” was influenced by various approaches including implementation of collaborative laboratory integrated reports (CLIR). As evident in Table 1, this single-tiered approach used by various state newborn screening programs within the USA resulted in extremely low positive predictive values (PPV) for MPS I.

This low PPV resulted from a combination of the choice of the cutoff for activity measurement positive calls, leading to carriers being identified and more importantly this low PPV relates to the significant incidence of pseudodeficiency of IDUA. Pseudodeficiency is defined as evidence of IDUA activity at levels seen in affected individuals yet with no evidence of a block in the catabolism of glycosaminoglycans (GAGs). Pseudodeficiency is felt to be related to the non-physiologic substrates (disaccharides) and assay conditions that are used to measure lysosomal enzyme activities. By definition, an individual who has pseudodeficiency of IDUA has no elevation of GAGs in tissues and therefore is not at risk of developing MPS I.

There is a significant impact of a low PPV on both families as well as health care providers, which should not be understated. Presented with a single-tier positive MPS I newborn screen result, primary care physicians lack the specialized training and knowledge with which to provide accurate and helpful information to a family even if information is available to them through web-based portals. Indeed, even inexperienced geneticists and metabolic specialists may not be well versed in the nuances of this finding and with good intention may attempt to resolve the issue with either further enzyme analysis, molecular genetic testing, or metabolic testing. This can be associated with a 4 to 6-week turnaround time, not to mention potential delays related to the complexity in obtaining insurance coverage for these analyses. The subsequent results may be difficult and confusing to interpret as IDUA enzyme activity will undoubtedly be confirmed to be low, and variants within the IDUA gene may be identified and classified as any combination of either variant of uncertain significance (VUS), pseudodeficiency or pathogenic.

Since the implementation of single-tier MPS I newborn screening in selected states within the USA, the National MPS Society has received 10–15 annual contacts from both distressed families and healthcare providers, who relate a mixed history including inaccurate and false information related to the interpretation of the newborn screen result, as well as interpretation of subsequent testing. This confusion leads to anger and distrust related of the entire process of MPS I newborn screening. Oftentimes, the National MPS Society provided comprehensive education to these contacts about genetic testing, newborn screening and pseudodeficiency, and provided emotional support and the understanding of newborn screening results to families that did not have clarity. This problematic situation, and the resulting confusion, has made it difficult and challenging for some families in whom MPS I is ultimately confirmed to accept the diagnosis and proceed with a treatment plan. In addition, families where the diagnosis is ultimately excluded can be left with a feeling of insecurity.

The incorporation of a second-tier biomarker to MPS I newborn screening has now been shown by a number of centers to reduce substantially the false positive rate of single-tier testing. Appropriately, the second tier consists of measurement of glycosaminoglycans within the dried blood spot card. Such a second-tier assay is a logical approach, as it encompasses the fundamental concept that an MPS I diagnosis requires demonstration of *both* deficiency of IDUA enzyme activity as well as elevation of the storage product, GAGs. By combining postanalytical interpretation using CLIRs and the addition of a GAG blood spot based second tier, Peck et al. increased the PPV from a cohort of 1213 specimens at risk for MPS I by single-tier screening to a PPV of 74% (17/23) [20]. Similar experience has been recently reported in Italy where retrospective incorporation of a second-tier GAG analysis resulted in a PPV of 100% [15].

## 3. Conclusions

The incorporation of a second-tier biomarker to MPS I newborn screening programs is an essential and necessary component to moving MPS newborn screening forward in the current era of precision medicine. As illustrated by Herbst et al. [21], there are multiple effective methods for GAG quantification from blood spot cards. These methods should immediately be integrated within current and future MPS newborn screening programs. There are multiple means by which this second-tier could be implemented, including direct integration by the screening lab itself or by creating regional second-tier testing facilities. Although the latter approach would increase the turnaround time to complete screening, the improved accuracy and predictive value would more than compensate for any delay. Since MPS I is not a rapidly progressive disorder, slight prolongation of testing times, by perhaps weeks, would not negatively influence patient’s ultimate outcomes. With such a high PPV, a screen positive patient can then be referred immediately and with confidence to a center that is experienced in the assessment and treatment of MPS individuals, and thus be promptly evaluated, receive accurate information and be initiated on an appropriate care path. An important component of this subsequent evaluation would include further biomarker studies, thorough clinical assessment and molecular genetic testing. Molecular genetic testing is best left to this stage as there is considerable allelic heterogeneity underlying MPS I and the molecular genetic information needs to be interpreted by a knowledgeable physician in context to the clinical and metabolic assessments in order to provide appropriate and accurate therapeutic options to the family [22]. The clinical evaluation should involve a multisystem assessment which together with the laboratory data will aid in the determination of the patients’ phenotype as discussed by Clarke et al. [23]. This assessment should include a general physical exam with particular emphasis on the presence of hernias, facial coarseness, hepatosplenomegaly, respiratory symptoms, altered joint range of motion and gibbus deformity. In addition, ophthalmologic evaluation to identify corneal clouding, cardiac evaluation including echocardiography to evaluate both valvular thickening and muscular function and a skeletal survey to evaluate dysostosis multiplex. Central nervous system imaging, initially with cranial ultrasound to assess for hydrocephalus and developmental evaluation, should also be considered. It is critical after confirmation of the diagnosis to determine if the patient fits either Hurler syndrome or attenuated MPS I, as this assignment will guide the management plan. This determination requires the integration of the clinical and molecular genetic findings and may not at this time be able to classify all patients, particularly less than the age of 4–6 weeks. The detection of cardiac, corneal, central nervous system or respiratory involvement and evidence of dysostosis multiplex in a newborn or young infant would be consistent with Hurler syndrome. Although the current recommendation for a patient diagnosed with Hurler syndrome is to offer early transplantation, there are no data that precisely delineate the most favorable time to initiate transplantation [24]. Currently, there are also insufficient published data to statistically evaluate the diagnostic utility of very early clinical findings in MPS I. It is hoped that as experience with NBS for MPS I increases, these clinical data will emerge.

Advancements and experience in MPS I newborn screening has quickly brought the field to a point where the addition of a second tier is necessary. We need to ensure that this second tier is quickly integrated so we do not turn the solution for a diagnostic odyssey provided by newborn screening into a confirmation odyssey.

## Figures and Tables

**Table 1 IJNS-06-00091-t001:** US Single-Tier Screening Program Data.

Site	Number Screened	Positive Call	Confirmed Dx	PPV
North Carolina [16]	62,734	19	1	5.3%
New York [17]	35,816	13	0	-
Illinois [18]	219,793	151	1 ^†^	0.7%
Missouri [19]	308,000	133	2 ^†^	1.5%

^†^: Excluded unresolved, indeterminant and lost to follow up cases.

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
