# Peer review of "Newborn Screening for Mucopolysaccharidosis I: Moving Forward Learning from Experience"

_2409-515X, 2020, doi:10.3390/ijns6040091_

Round 1

Reviewer 1 Report

This is a succinct, convincingly argued commentary/review on the advantages of second tier testing in newborn screening for MPS I. The paper is well referenced and well written.

I found only one issue that requires further information: In Lines 125-126: The authors recommend positive cases be referred to centres experienced in assessing and managing patients with MPS disorders for “further biomarker studies, thorough clinical assessment and molecular genetic testing”  Although they address biomarkers and molecular genetic testing elsewhere in the paper, they provide no supporting information on “a thorough clinical assessment”. Some comment, preferably supported by data, on the diagnostic utility of a clinical assessment in this cohort at such a young age, but with a high a priori risk of being affected, would be both useful and educational.

2 typos:

Line 32: experimental

Line 121 – patients’

Author Response

The reviewers comments are quit helpful, our responses are:

1) the word experiential is appropriate (meaning related to experience)

2) Typo corrected

3) We have addressed this important comment by adding the following paragraph to the manuscript with new references.

"The clinical evaluation should involve a multisystem assessment which together with the laboratory data will aid in the determination of the patients’ phenotype as discussed by Clarke et al. [23].  This assessment should include a general physical exam with particular emphasis on the presence of hernias, facial coarseness, hepatosplenomegaly, respiratory symptoms, altered joint range of motion and gibbus deformity.  In addition, ophthalmologic evaluation to identify corneal clouding, cardiac evaluation including echocardiography to evaluate both valvular thickening and muscular function, and a skeletal survey to evaluate dysostosis multiplex.  Central nervous system imaging, initially with cranial ultrasound to assess for hydrocephalus and developmental evaluation should also be considered.  It is critical after confirmation of the diagnosis to determine if the patient fits either Hurler syndrome or attenuated MPS I, as this assignment will guide the management plan.

This determination requires the integration of the clinical and molecular genetic findings and may not at this time be able to classify all patients, particularly less than the age of 4-6 weeks..  The detection of cardiac, corneal, central nervous system or respiratory involvement and evidence of dysostosis multiplex in a newborn or young infant would be consistent with Hurler syndrome.  Although the current recommendation for a patient diagnosed with Hurler syndrome is to offer early transplantation, there are no data that precisely delineates the most favorable time to initiate transplantation [24].  Currently there are also insufficient published data to statistically evaluate the diagnostic utility of very early clinical findings in MPS I.  It is hoped that as experience with NBS for MPS I increases these clinical data will emerge."     

Reviewer 2 Report

A limitation of newborn screening that is being implemented for MPS-1 is low positive predictive value of the 1st tier screening. This commentary recommends incorporation of second-tier biomarker to MPS-1 newborn screening programs. The content of this manuscript is based on the front-line information on the limitation of the current NBS program for MPS-1 as well as potential approaches as the second-tier biomarker. The manuscript is organized well and can be shared by all medical professionals that involve in the MPS-1 newborn screening programs.

Author Response

We thank the reviewer.